# XENOFOOD—An Autoclaved Feed Supplement Containing Autoclavable Antimicrobial Peptides—Exerts Anticoccidial GI Activity, and Causes Bursa Enlargement, but Has No Detectable Harmful Effects in Broiler Cockerels despite In Vitro Detectable Cytotoxicity on LHM Cells

**DOI:** 10.3390/pathogens12030458

**Published:** 2023-03-14

**Authors:** András Fodor, Tibor Vellai, Claudia Hess, László Makrai, Károly Dublecz, László Pál, Andor Molnár, Michael G. Klein, Eustachio Tarasco, Sándor Józsa, Petra Ganas, Michael Hess

**Affiliations:** 1Department of Genetics, Eötvös Loránd University, Pázmány P. Sétány 1C, H-1117 Budapest, Hungary; vellai.tibor@ttk.elte.hu; 2Clinic for Poultry and Fish Medicine, Department for Farm Animals and Veterinary Public Health, University of Veterinary Medicine (Vetmeduni Vienna), 1210 Vienna, Austria; claudia.hess@vetmeduni.ac.at (C.H.); petra.ganas@bfr.bund.de (P.G.) michael.hess@vetmeduni.ac.at (M.H.); 3Department of Microbiology and Infectious Diseases, University of Veterinary Medicine, H-1581 Budapest, Hungary; 4Institute of Physiology and Nutrition, Georgikon Campus, Hungarian University of Agriculture and Life Sciences (MATE), Deák Ferenc utca 16, H-8360 Keszthely, Hungary; dublecz.karoly@uni-mate.hu (K.D.); pal.laszlo@uni-mate.hu (L.P.); info.mobildoki@gmail.com (A.M.); jozsa.sandr@gmail.com (S.J.); 5USDA-ARS & Department of Entomology, The Ohio State University, 13416 Claremont Ave., Cleveland, OH 44130, USA; klein.10@osu.edu; 6Department of Soil, Plant and Food Sciences, University of Bari “Aldo Moro”, Via Amendola 165/A, 70126 Bari, Italy; eustachio.tarasco@uniba.it

**Keywords:** multidrug resistance, MDR, EPB (entomo-pathogenic bacteria, *Xenorhabdus*, (*X. budapestensis*, *X. szentirmaii*, *X. innexii*), and *Photorhabdus* species), CFCM (cell-free conditioned media), NR-AMP (non-ribosomal-templated antimicrobial peptides), anti-microbial, anti-bacterial, anti-coccidial, anti-protist activity, cytotoxicity, *Clostridium* CFU—(colony forming unit), in situ (local) bioavailability, XENOFOOD, alternative antibiotics

## Abstract

Entomopathogenic bacteria are obligate symbionts of entomopathogenic nematode (EPN) species. These bacteria biosynthesize and release non-ribosomal-templated hybrid peptides (NR-AMPs), with strong, and large-spectral antimicrobial potential, capable of inactivating pathogens belonging to different prokaryote, and eukaryote taxa. The cell-free conditioned culture media (CFCM) of *Xenorhabdus budapestensis* and *X. szentirmaii* efficiently inactivate poultry pathogens like *Clostridium*, *Histomonas*, and *Eimeria*. To learn whether a bio-preparation containing antimicrobial peptides of *Xenorhabdus* origin with accompanying (in vitro detectable) cytotoxic effects could be considered a safely applicable preventive feed supplement, we conducted a 42-day feeding experiment on freshly hatched broiler cockerels. XENOFOOD (containing autoclaved *X. budapestensis*, and *X. szentirmaii* cultures developed on chicken food) were consumed by the birds. The XENOFOOD exerted detectable gastrointestinal (GI) activity (reducing the numbers of the colony-forming *Clostridium perfringens* units in the lower jejunum. No animal was lost in the experiment. Neither the body weight, growth rate, feed-conversion ratio, nor organ-weight data differed between the control (C) and treated (T) groups, indicating that the XENOFOOD diet did not result in any detectable adverse effects. We suppose that the parameters indicating a moderate enlargement of bursas of Fabricius (average weight, size, and individual bursa/spleen weight-ratios) in the XENOFOOD-fed group must be an indirect indication that the bursa-controlled humoral immune system neutralized the cytotoxic ingredients of the XENOFOOD in the blood, not allowing to reach their critical cytotoxic concentration in the sensitive tissues.

## 1. Introduction

The emergence, and re-emergence of diseases caused by multidrug-resistant (MDR) pathogens, and parasitic organisms in plants, invertebrates, vertebrate animals, and humans provide arguments for the urgent search for new antimicrobial-active drugs with novel modes of action [1,2,3,4,5,6,7,8,9,10,11,12]. This study intended to contribute to the research field on the drug potential of natural biosynthetic antimicrobial peptides as chemotherapeutic tools against multi-drug resistant pathogens, especially, multi-drug resistant pathogens, [13]. The term “antimicrobial peptides” (AMPs) [14] includes any polyamide (or even biopolymer with ester, thioester, or otherwise modified backbone) that can be made on a contemporary chemical peptide synthesizer. The limit in size is greater than the arbitrary cutoff of 50 amino acids set up by the US Food and Drug Administration [15] for proteins and far exceeds that of biological recognition elements [14]. Therefore, not only the gene-encoded, ribosomal templated antimicrobial peptides, (RT-AMPs) but other peptides of antimicrobial activity, including enzymatically (bio)-synthesized non-ribosomal templated antimicrobial peptide (NR-AMP) molecules should also be considered. The AMPs are of great perspectives to combat MDR prokaryotes because antibiotic-resistant bacteria perform a high frequency of collateral sensitivity to antimicrobial peptides, [16]. Furthermore, the mobility patterns of AMP-resistance genes differ from those of the antibiotics-resistance genes [17]. AMPs are produced by all but Archea taxa, [18], and are considered innate components of the innate immune systems of all known eukaryotic organisms, [19]. The RT-AMPs are usually narrow target spectral drugs, [20], but their target spectrum can be extended in vivo [21] probably due to their immune-modulatory actions [22,23]. The “secrete” of their future drug potential is their molecular versatility, allowing QSAR modeling [24], and computer-aided design of antimicrobial peptide analogs, and peptides [18,25], and benefiting from the well-suited and powerful tool of proteomics, for a better understanding the respective study molecular responses to antimicrobial compounds, [26]. The vast majority of MPs are membrane-active [25] Others penetrate and bind to intracellular targets like proline-rich PrAMPs, [27]. Several AMPs proved efficient against pathogenic-parasitic *Leishmania* and *Trypanosoma* species. Unlike those of the RT-AMPs, the target spectrum of most NRP-AMPs is usually large [28,29]. In eukaryotic targets, some induce apoptotic death of leishmanial protist parasites through a calcium-dependent, caspase-independent mitochondrial toxicity mechanism [30].

Entomopathogenic bacteria (abbreviation: EPBs), the obligate symbionts of entomopathogenic nematode (EPN) species synthesize and release NRP-hybrid peptides that provide well-balanced pathobiome conditions for the respective EPN/EPB symbiotic complex in polyxenic (insect cadaver in the soil) environments [31,32]. These bacteria are considered potential sources of potent natural anti-microbial [31], and anti-protist [33] compounds. We have previously found extreme antimicrobial-active secreted peptides produce by *Xenorhabdus budapestensis* (EMA) and *X. szentirmaii* (EMC) [34,35] in both solid and liquid media that were active against Gram-positive and -negative pathogenic bacteria, [34,36]; anti-plant pathogenic bacterial [37,38], anti-coccidial [39], anti-oomycetes [37,40], as well as antifungal (Ujszegi et al., in preparation) activity. ENMA and ENC are not simply good but the best antimicrobial-peptide-producing *Xenorhabdus* strains we have ever seen so far.

As for the latest exciting trend of this research field, it has recently been discovered that the operons encoding the various enzymes of the biosynthetic pathway of each NR-AMP are globally regulated by the gene called *Hfq* in EPB species, providing an option to create EPB strains each of which produces only one single NR AMP molecule. (The method of constructing such a strain in the lab is called by using the “easyPACId” method) [41].

The main aim of this study was to learn whether our per os administered, food-supplement bio-preparation (called XENOFOOD) which contains antimicrobial peptides of *Xenorhabdus* origin (with accompanying in vitro detectable) cytotoxic effects) as ingredients could be considered a safely applicable preventive feed supplement.

We conducted a 42-day feeding experiment on freshly hatched broiler cockerels. XENOFOOD (containing autoclaved *X. budapestensis*, and *X. szentirmaii* cultures developed on chicken food) were consumed by the birds. The XENOFOOD exerted detectable gastrointestinal (GI) activity (reducing the numbers of the colony-forming *Clostridium perfringens* units in the lower jejunum. No animal was lost in the experiment. Neither the body weight, growth rate, feed-conversion ratio, nor organ-weight data differed between the control (C) and treated (T) groups, indicating that the XENOFOOD diet did not result in any detectable adverse effects. That is in vitro toxic ingredients of the XENOFOOD seemed to be inactivated in vivo in chicken.

We suppose that the parameters indicating a moderate enlargement of bursas of Fabricius (average weight, size, and individual bursa/spleen weight-ratios) in the XENOFOOD-fed group must be an indirect indication that the bursa-controlled humoral immune system neutralized the cytotoxic ingredients of the XENOFOOD in the blood, not allowing to reach their critical cytotoxic concentration in the sensitive tissues.

This conclusion may have some significance concerning the application perspective. Whether the different NR-AMPs could ever be utilized as drugs not only against prokaryotic (bacteria) pathogens but eukaryotic (fungal pathogens, and parasitic protists) depends on the side effects. To get experimental experience with the option of applying EPB-produced antimicrobials to pathogens, and parasites of veterinary significance, we present here the results of an in vitro, and an accompanying in vivo study on chicken. In the in vitro study, we tested the cytotoxic potential of the cell-free conditioned culture media (CFCM) of three entomopathogenic bacterium species, —*X. budapestensis*, DSM16342 (EMA); *X. szentirmaii* DSM16338 (EMC); *Photorhabdus luminescens* ssp. *akhurstii* TT01-on chicken tissue culture cells, namely, on the Leghorn Male Hepatoma (LMH, see Materials and methods) Icells, (a permanent confluent hepato-carcinoma cell line). Each CFCM proved rather cytotoxic in this test. In the in vivo study, we fed freshly hatched male broiler chickens for 42 days with XENOFOOD [39] which contained autoclaved cultures of EMA, and EMC. These bacteria were grown on standard chicken (starter and grower) feed, and the whole culture was used as a “feed supplement”. It had been known that these EPB species cannot grow at body temperature (above 33 °C).

## 2. Results

### 2.1. Anti-Clostridial Activity of EMA and EMC In Vitro

Based on the previous results of repeated experiments, we choose *X. budapestensis* DSM16342 (EMA) and *X. szentirmaii* DSM 16338, (EMC) which had been identified in our laboratory, [34] and seem to be the most promising strains as antimicrobial producers we have ever come across so far. This statement also stands for their anti-Clostridial potential. For the exact description and history of these strains, see Ref. [35]. We retested the in vitro efficiency [36,37,38] in overlay bioassays and the pepsin resistance of the active ingredient that we previously published [39], (Figure 1a,b).

Both native and autoclaved CFCM samples of EMA and EMC bacterial strains were bioassayed both in liquid, and agar diffusion tests as described previously [36,37,38,39]. The anticoccidial activities of native and pepsin-treated CFCM samples were compared similarly as described previously [39]. All previous data on antimicrobial activity against each of the tested substances on different clinical isolates of Gram-positive *Staphylococcus aureus* [35,36,37,38,39], Ref. [42] was repeatedly confirmed, indicating the antimicrobial activity These data were especially important as confirming all previous findings published in [38]. The experiment also confirmed the thermostability and the pepsin durability of the CFCM of EMA, and EMC [40], since autoclaving of these CFCMs did not affect the antimicrobial activity against any of the tested organisms. 

### 2.2. In Vitro Bioassays: Cytopathogenic Effects of Cell-Free Condition Media (CFCM) of 3 EPB Species on LMH Chicken Cell Monolayer

Two experiments were performed on this subject, (see Section 4). In both experiments, permanent, confluent monolayer culture samples (developed in different Falcon flasks) were used as target organisms, and different dilutions of different CFCMs were tested.

(As described in the Materials and Methods, confluent monolayers of LMH cells were parallelly developed in 18 Falcon flasks (for the first experiment), and 36 flasks (for the second experiment). The culture media were Medium RPMI 1640 supplemented with 10% FBS, penicillin G (200 IU/mL), and streptomycin (200 µg/mL). The incubation time was 72 h in both cases, in the controlled atmosphere of 5% CO_2_ at 37 °C and around 85–90% humidity in 25 cm (2) flasks with filtered caps. After the layer formed, the liquid growth media was removed from all but 3 flasks (in the first experiment) and in 2 × 2 flasks (in the second experiment). Those few (3 × 1 and 1 × 2) flasks were referred to as “unchanged” control flasks. As for other flasks, 3 × 1 (in the first experiment) and 1 × 2 (in the second experiment) were refilled e fresh culture media (referred to as M199 + 15% FKS control flasks) while the rest of the flasks were refilled 100 *v*/*v*% of different CFCMs (EMA, EMC, TT01 yellow and TT01 red, in the first experiment, and different dilutions of EMA and EMC CFCMs in the second experiments. As for the replicates, in the first experiment we worked with triplicates, and in the second experiment with duplicates).

#### 2.2.1. Experiment-1

The aim was to compare the cytopathogenic effects of 4 different undiluted CFCMs obtained from cultures of EMA, EMC, and TT01 yellow, and TT01 red colony color variants, respectively, (see Section 4). It was found that each CFCM exerted destructive effects on the cell layer. The initial lesions on the LMH monolayer were followed by more serious ones consisting of clumps attached to the monolayer, (Score 3).

The results (data given as empiric score values, (see Section 4) of the Experiment are presented in Table 1.

We interpret these results as experimental proof of the presence of strong cytotoxic molecules in each of the 4 CFCMs. The four CFCMs differed in their cytotoxicity, the order of toxicity was: EMA > EMC > T0T1. Considering that the CFCMs filtrates contain not only AMPs but toxin protein molecules as well, we should know whether the experienced cytotoxicity was (a) a side effect(s) of the NR-AMP(s), or/and the toxin(s), and our results could not answer this question. Another question that remained open is whether the absence of any sign of in vivo toxicity was due to the denaturation of the toxic (protein-like) ingredients of the denatured CFCMs. We saw the antimicrobial activity of the CFCMs did not disappear after being autoclaved. There are the following experimental conclusions that can be drawn: (1) These experiments should be repeated with the peptide-rich fraction [38] and/or separated single NR-AMP molecules like (first of all) the fabclavines [43,44,45,46].

#### 2.2.2. Experiment-2

The aim was to compare the cytopathogenic effects of serial dilutions of the CFCM filtrates of EMA, and EMC cultures, considering that both EMA and EMC CFCMs showed much stronger antimicrobial activity than the two TT01 CFCMs. The stock solutions, (and serial dilutions of them) used in this experiment were: EMA 60%, and EMC 80%, respectively. Confluent monolayer from LMH cells was developed again in Medium RPMI 1640 supplemented with 10% FBS, penicillin G (200 IU/mL), and streptomycin (200 µg/mL) for 72 h in a controlled atmosphere of 5% CO_2_ at 37 °C and around 85–90% humidity in 25 cm^2^ Falcon flasks with filtered caps (see also in Materials, and Methods). The same score system [47] (Amin, 2012, see Appendix A) was used for evaluating the results. The results of Experiment 2 are presented in Figure 2 and Figure 3, and the data and statistics are given in Appendix A.

The experimental conclusion is that the cytotoxic effect of the CFCM of the EMA was not only severe but unambiguously dose-dependent throughout the experiments. The histological pictures of replicates seemed practically uniform.

The differences between the values represented by the 1st (orange) columns and the rest of them proved significant at (0.001 < *p* < 0.005 level) at each consecutive measurement. The replicate histological pictures seemed practically uniform, and below 12 *v*/*v*% the cytotoxic effect of the CFCM of the EMC was rather moderate, and not dose-dependent.

We interpret these results as experimental proof that the in vivo application either of EMA NR_AMP peptides or the EMC NR_AMP peptides did cause fatal or even detectable cytopathogenic in sensitive tissues (like liver cells) in a dose-dependent manner. The degree of destruction was serious and irreversible above 3.2 *v*/*v*% (1:25 dilution) EMA-CFCM and above 2.4 *v*/*v*% EMC. The experimental conclusion is that in each of both cases, there was a threshold concentration below which no cytotoxic destruction is observable. Consequently, if an antimicrobial compound with cytotoxic side effects (like an NR-AMP) had been orally administered in a higher dose, which is needed for its antimicrobial efficacy in the gut, but then reached its critical cytotoxic threshold concentration neither in the blood nor in any sensitives tissue (because one or another reason), then in vivo *toxicity* would not be detected. 

### 2.3. Results of the In Vivo Xenofood-Feeding Experiment

#### 2.3.1. Gastrointestinal Activity of XENOFOOD

To examine whether the gastrointestinal potential of the XENOFOOD was considerable we counted *Clostridium* CFUs in the lower ileum (see Figure 4) of XENOFOOD-fed and untreated animals. 

Using ANOVA statistical analysis, we compared the average of the lower ileal *Clostridium* CFUs (see Figure 2) of the XENOFOOD-fed animals (48.1) with those of the controls (149.9). Although we found that despite the 3X difference, only a slightly significant could be demonstrated, probably due to the large standard deviation in the control values (Table 2). and the very low infestation rate in the animals chosen for the experiment during the 42-day long experiment. In other terms, only very few animals had been infected with *Clostridium*.

We found that despite the 3X difference, only a slightly significant difference could be demonstrated, probably due to the large standard deviation in the control values (Table 3), and because of the very low infestation rate in the animals chosen for the experiment.

The data show that the average weights of the spleens were smaller, while the average measurements of the sizes of the bursas were larger in the XENOFOOD-fed group, and the differences were significant. The same can be stated about the *Clostridium* colony-forming units, (CFUs): there were fewer in the XENOFOOD-fed group (on average). The XENOFOOD exerted unambiguously detectable gastrointestinal antimicrobial activity. But the original infestation of the population from which the experimental animals were selected rate was very low. These results allow concluding that XENOFOOD feeding was harmless. We suppose that the bursa-dependent humoral immune system must have neutralized the cytotoxic peptide-like molecules (including the GI-active NR-APMPs) of the XENOFOOD in the blood, protecting the sensitive tissues and organs in such a way.

#### 2.3.2. Growth Rate, Feed Consumption, Feed Conversion, and the Post-Mortal Data

During the 42-day-long experiment, no animal was lost. We did not detect any significant differences either in the feed intake, body weight gain, and fed conversion ratio (FCR) between the control and XENOFOOD-fed groups (Table 4).

As for the organ weight, they followed the allometric rules, according to which the regular and systematic patterns of growth such as the mass or size of any organ or part of a body can be expressed about the total mass or size of the entire organism according to the allometric equation: Y = bxα, where Y = mass of the organ, x = mass of the organism, α = growth coefficient of the organ, and b = a constant (only bursa- and spleen data are given). This fact itself excludes any organ-development abnormality of the treated (XENOFOOD-fed) animals. But the larger bursa weight and the smaller spleen weights, as well as larger bursa/spleen ratios in the XENOFOOD-fed groups may indicate indirectly an intensified humoral immune activity, leading to the neutralization of the cytotoxic AMP-components in the blood (Table 4).

Summarizing the results of the in vivo feeding experiment, their interpretation is as follows: we wanted to learn two things: (1) whether the per os administered XENOFOOD (1) has a preventive potential against *Clostridium* infection; (2) can be considered as a harmless or as a risky food supplement. The experimental conclusions are, that (1) XENOFOOD must have a preventive potential but the experiment should be repeated with experimentally infested birds, where infection inoculum is strong and standardized. (2) The XENOFOOD proved a harmless food supplement, and despite the in vitro cytotoxicity of its CFCM ingredients, did not cause any detectable adverse effect in the animals of the XENOFOOD-fed (T) groups of the birds.

## 3. Discussion

When considering the potential benefits of studying and discovering AMPs of EPN/EPB symbioses we must remind of the natural role of these compounds. There are several biosynthetic (NR-AMPs) produced by *Xenorhabdus* species. They are end-products of nonribosomal peptide synthetases (NRPSs) which usually use terminal reductase domains for 2-electron reduction of the enzyme-bound thioester releasing the generated peptides as C-terminal aldehydes [49]. Each EPN/EPB symbiotic association has been, existing in a given environmental milieu, as a coevolutionary product selected by/in a respective polyxenic environment (insect cadaver in the given soil). The biological role of each NR-AMP molecule is a contribution to providing pathobiom) conditions for a given symbiotic complex [32]. We can give only a short list of those NR_AMPs discovered in our *Xenorhabdus* bacteria so far without having information about their modes of action. Structure elucidation of natural products including the absolute configuration is a complex task that involves different analytical methods like mass spectrometry, NMR spectroscopy, and chemical derivation, which are usually performed after the isolation of the compound of interest [50]. As for *X. budapestensis* DSM16342T(Nov) (EMA), in our previous experiments, we found an unusual arginine-rich hexapeptide bicornutin [37] which we mistakenly believed to be the active ingredient of the CFCM of EMA species, because it has always been present in the antimicrobial-active CFCM [37]. Our results drew the attention of Professor H. Bode and his associates. Continuing our search for strong AMPs in our EMA Fuchs and his associates (2012), proved that bicornutin exerted neither bacteriostatic nor bactericide activity, however [43], but this work initiated further search partly toward this strange molecule (studying the neutral loss fragmentation-pattern-based screening for arginine-rich natural products. They extended their studies on 16 different *Xenorhabdus* and *Photorhabdus* strains) and toward the substantive antibiotic component of the CFCM (the fabclavine). The bicornutin research led to the discovery of a novel group of related natural products comprising 25 different arginine-rich peptides. These molecules were identified due to their characteristic neutral loss fragmentation pattern, and the structures of eight of these compounds were elucidated. Two biosynthesis gene clusters encoding non-ribosomal peptide synthetases were also identified, emphasizing the possibility to identify a group of structurally and biosynthetically related natural products, [43]. The research was expanded by the same fellow scientists, leading to the discovery of the fabclavine [44] which seems to be the most significant AMP in the genus *Xenorhabdus*, and EMA and EMC proved the most efficient fabclavine-producers. Fabclavins, which were discovered in EMA and EMC seem to be the most important NR_AMP molecular family in the genus *Xenorhabdus* [45,46,51]. EMA and EMC seem to be the strongest fabclavine producers [35]. Surprisingly enough *X. budapestensis* which has been discovered in samples of Central European (Hungarian) soil has also been found in two different locations in China, and produce another type of NR-AMPs, two novel cyclic depsipeptides Xenematides F and G. but no one tested for fabclavine production this new isolate [52,53]. As for *X. szentirmaii*, it produces xenofuranones A and B [54], and phenazines [55], and szentiamide, [56,57].

The coexistence of so many antimicrobial compounds with large target spectra and cooperating potential comprises a potentially promising toolkit combatting MDR because antibiotic-resistant bacteria perform a high frequency of collateral sensitivity to antimicrobial peptides, [13,16,17]. *X. budapestensis* (EMA) and *X. szentirmaii* (EMC) are abundant sources of NR-AMP molecules of large target spectra and strong anti-pathogen potential, [35].

Each NR-AMP molecule in the *Xenorhabdus* CFCM is the end-product of a biosynthetic pathway. Each biosynthetic pathway is encoded by genes clustered in the respective operon (biosynthetic gene complex, BCG). In EMC 71 BCGs have been identified [58]. Their large spectral NR-AMP products of overlapping activities and cooperating potential are coexisting as ingredients of the CFCM. Theoretically, there are two alternative ways of research philosophy for the geneticist of agricultural or veterinary commitments.

One option is to focus on searching for individual molecules as new potential drugs, benefitting the global (co)regulation of the biosynthesis of different NR_AMPs, searching for or constructing EPB strains synthesizing only single NR-AMP molecules. As for this alternative, the recent discovery, according to which the different NRPS-encoding operons are under the global control of the post-transcriptional regulator, *Hfq*, provides an option for constructing *Xenorhabdus* strains producing single NR-AMP only [41]. It is a great challenge!

Another strategy is to try to benefit from the co-existence of naturally cooperating NR_AMP molecules in the CFCMs by producing harmless but effective preventive biopreparations like food supplements either as food supplements for veterinary, or soil additives for plant protection goals against pathogens and parasites.

This latter option may be discussed here in light of the experiment’s results where chickens were fed by XENOFOOD, a food supplement, in which in vitro active ut heat-sterilized NR-AMPs were present as ingredients. We know about these compounds that they were of peptides [28], in vitro anti-coccidial active [39], which did not lose their antimicrobial potential either when heat-sterilized [36], or when subjected to pepsin digestion [39].

But in the in vitro experiment we presented here we found that each EPB CCM we tested exerted strong cytopathogenic effects on LMH permanent confluent chicken cells. We do not know whether one single (most probably the fabclavine) [43,44,45,46] molecule or more than one molecule can be taken as responsible for the cyto-pathogenecity. There are several candidates (see Table 5).

It is not known whether the mechanism of the antimicrobial action of a given NR-AMP molecule is the same as those causing cytopathogenicity in the organisms.

The XENOFOOD food supplement proved in this experiment harmless and GI active against a bacterium pathogen, *Clostridium perfringens*.

However, in our in vivo feeding experiment the lower number of *Clostridia* does not automatically indicate a “protection” exerted by the XENOFOOD, since the natural infestation rate of the experimental animals was very low, making conclusions on the preventive potential of the XENOFOOD just hypothetical, probable, but not certain. Another plausible explanation however for the strong differences between the average CFU values between the control (C) and the XENOFOOD-fed (T) group cannot be given.

The moderate but detectable anti-Clostridial effect without any detectable harmful side effects may be explained in more ways than one. An explanation may come from the previous experiments on *Eimeria* (unpublished) that the effective cytotoxic concentration is much higher than the effective anti-clostridial dose.

An alternative interpretation is that the actual local NP-AMR concentration (bioavailability) was much lower in the blood than in the GI system. Consequently, the local critical threshold cytotoxic concentration of the toxic compound(s) in other organs and tissues was not reached.

We suppose that the two alternatives cannot exclude each other but that the second option is more plausible because the immune system of the birds must be capable of neutralizing peptide-like compounds, either the toxins or the NR-AMPs or both. The higher average weights of the bursas in the T-groups suggest that humoral immunity is mainly responsible for that. 

Perspectives are discussed in the Section 5 “Conclusions”.

## 4. Materials and Methods

The anti-clostridial potential of EPB CFCM: The respective methods have been published in [39]. Briefly, *Clostridium perfringens* NCAIM 1417 strain was obtained from the National Collection of Agricultural and Industrial Microorganisms–WIPO (of Hungary, Faculty of Food Sciences, Somlói út 14–16 1118 Budapest, Hungary). *Clostridium perfringens* LH1-LH8; LH11-LH16; LH19, and LH20 are of chicken origin, and LH24 came from a pig; each has been deposited in the (frozen) stock collection of the Department of Microbiology and Infectious Diseases, University of Veterinary Medicine Budapest, Hungary. *Xenorhabdus* strains, *X. budapestensis* DSM 16342(T) (Lengyel) (EMA), and *X. szentirmaii* DSM 16338 (T) (Lengyel) (EMC) [34,35]. of EMA and EMC CFCM bioassays were tested on different Gram-positive strains including *Clostridium perfringens* strains were carried out as described before [35,36,37,38,39]. In vitro experiments: Cell-free conditioned culture media (CFCM), of antibiotic-producing bacteria, were tested, and LHM (tissue culture) cells were in two different experiments.

*Experiment 1:* Testing cytotoxicity of different EPB CFCM on confluent LMH (leghorn male hepatoma, LMH) cell line [48]: *Developing Confluent layers of LMH (leghorn male hepatoma, LMH) cell line* [48]: (LMH; ATCC Number: CRL-2117™) were developed culture Falcon flasks in Medium RPMI 1640 (Invitrogen/GIBCO), supplemented with 10% heat-inactivated fetal bovine serum FBS (Invitrogen/GIBCO), penicillin G (200 IU/mL) and streptomycin (200 µg/mL), respectively.

In detail, cells were inoculated into 25 cm (2) flasks with filtered caps (Sarstedt) containing an end volume of 7 mL culture and incubated in a controlled atmosphere of 5% CO2 at 37 °C and around 85-90% humidity. After 72 h of incubation, a confluent monolayer of LMH cells was obtained per flask.

Altogether 18 Falcon tubes, —each containing 9 mL Medium 199 with Earle’s Salts, L-glutamine, 25 mM HEPES, and L-amino acids (Invitrogen/GIBCO) and supplemented with 15% heat-inactivated fetal bovine serum FBS (Invitrogen/GIBCO) and 0.22% rice starch were used, 2x3 controls and 4X3 experimental tubes. 

Preparation of CFCMs: Tubes with the same media (except for streptomycin, were inoculated with 4 different bacterial strains, EMA, EMC, TT01 yellow, or TT01 red, representing 2 *Xenorhabdus*, (*X. budapestensis* nov. DSM16342(T), Lengyel) (EMA), [34,35], and *X. szentirmaii* nov. DSM16338 (T) (Lengyel) (EMC, [34,35], and 1 *Photorhabdus* (*P. luminescens* ssp. *akhurstii* TT01, [75] species. The latter is obtained from the Boemare laboratory (Montpellier, France). TT01 yellow and TT01 red names were used for two colony-color variants segregating spontaneously in McConkey agar plates, (P. Ganas, unpublished). (*Xenorabdus* and *Photorhabdus* are penicillin-resistant species).

These antibiotic-producing bacteria were freshly taken from frozen cultures and grown on the bacterial species grown on MacConkey agar plates before they were transferred to the liquid medium as described before, [35,36,37,38,39,40]. The bacterial species were grown on MacConkey agar plates before they were transferred to the liquid medium, and unexpectedly two different types of colonies for TT01 were observed on the agar plates: red-brown colored colonies which adsorbed the neutral red from the MacConkey agar and yellow-colored colonies which did not so. Both types of colonies were tested for the effect of cell-free filtrates on LMH monolayers. In this particular experiment, the antibiotic-producing bacteria were cultured at 28 °C. The bacteria were incubated for 65 h at 30 °C in a shaker (225 rpm. Bacterial cultures were then centrifuged at 3300× *g* for 5 min and then the supernatants from the cultures were filtered through 0.22 µm cellulose acetate filters (Millipore).

*Experimental design*: From all but 3 of the Falcon flasks (with the 72-h old LMH-layers), the culture medium of the LMH monolayers was removed from the flasks and replaced by the 4 CFCMs. Each of the CFCMs was tested in triplicates. There were two sets of controls. There were also 3 flasks with fresh medium (fresh Medium 199 supplemented with 15% FBS) without CFCM, and another 3 with the original, (“unchanged”, that is 72 h old) culture media. All these 4 × 3 experimental and 2 × 3 control flasks were incubated for another 72 h. The cultures were incubated in a controlled atmosphere of 5% CO_2_ at 37 °C and around 85–90% humidity.

In vitro *Experiment 2*: permanent chicken liver cells (LMH (leghorn male hepatoma, LMH) cell line [48]. LMH; ATCC Number: CRL-2117™) [48] were grown in Medium RPMI 1640 (Invitrogen/GIBCO) supplemented with 10% heat-inactivated fetal bovine serum FBS (Invitrogen/GIBCO), penicillin G (200 IU/mL) and streptomycin (200 µg/mL). Cells were inoculated into 25 cm^2^ flasks with filtered caps (Sarstedt) containing an end volume of 7 mL culture and incubated in a controlled atmosphere of 5% CO_2_ at 37 °C and around 85–90% humidity.

After 72 h of incubation, a confluent monolayer of LMH cells was obtained per flask. All but the so-called “Unchanged culture flasks” LMH monolayers, the culture medium of the LMH monolayers was removed from the flasks and replaced by *something*. The control flasks were refilled with fresh media (Medium RPMI 1640 (Invitrogen/GIBCO) supplemented with 10% heat-inactivated fetal bovine serum FBS (Invitrogen/GIBCO), penicillin G (200 IU/mL) and streptomycin (200 µg/mL). The experimental flasks were refilled with cell-free (centrifuged and filtered serially diluted stock solutions (EMA: stock solution: 60 *v*/*v*; EMC stock solution 80 *v*/*v*%). The dilutions were 1:2.5; 1:5; 1:10 1:.25; 1:.50; 1:75, and 1:100, respectively. The EPB cells had been also cultured in Medium RPMI 1640 supplemented with 10% FBS, penicillin G, without streptomycin, (*Xenorhabdus* are penicillin-resistant). Each of the different filtrate analyses and the controls was performed in duplicate. The cultures were incubated in a controlled atmosphere of 5% CO_2_ at 37 °C and around 85–90% humidity.

Each monolayer was investigated visually by an inverted light microscope to detect the effect of the cell-free filtrates on LMH monolayers. According to the degree of monolayer destruction, the following scoring system was established and also published [47], (See Appendix A).

*XENOFOOD preparation:* XENOFOOD contained 5% soy-meal, which had been suspended with an equal amount (*w*/*w*) of EMA and another 5% suspended in an equal amount (*w*/*w*) of EMC cells obtained from 5 days-old shaken (2000 rpm) liquid cultures by high-speed (Sorwall; for 30 min) centrifugation. The liquid cultures were in 2XLB; (DIFCO); supplemented with meat extract equivalent amount to the yeast extract. The 5 days had been proven optimal for antimicrobial substance production at 25 °C in these conditions). It had previously been discovered that both EMA and EMC grow and produce antimicrobial substances in autoclaved soy meal containing some water and yeast extract or autoclaved yeast, (in 0.5 *w*/*w*%). Therefore, the original chicken food served as a semi-solid culture media of *Xenorhabdus* cells. Both EMA and EMC culturing semi-solid chicken food, which was prepared daily, and have been incubated in sterile conditions for another five days; then the EMA and EMC culturing media were united; autoclaved (20 min, 121 °C), and then dried by heat overnight. The *Xenorhabdus* cells were killed in such a way, while the heat stabile [36] antimicrobial compounds remained active.

*XENOFOOD* in vivo *feeding experiment: Experimental animals*: One-day-old male broiler chickens (N = 2 × 34 = 68) were equally distributed into two groups: Control (C) and Treated (T) groups. The latter was fed with XENOFOOD. The C group was kept on a normal starter (1–10-d), grower (11–24-d), and finisher (25–42-d) diet according to the standard international protocol, (permission number: PE/EA/3340-6-2016) [1]) given to (coauthor) Professor László Fodor as a Head of the Department of Microbiology and Infectious Diseases, University of Veterinary Medicine, H-1581, Budapest, P.O. Box 22, Hungary, [in Hungarian]). The animal experiment was a cooperative experiment of the Fodor Department, and the Dublecz Department for practical technical reasons. The composition and nutrient content of the experimental diets can be found in Appendix A. During the rearing period feed intake, growth rate, and feed conversion of animals were measured on a pen basis for the starter, grower, and finisher phases.

*Dissection, post-mortal data:* Not all but a sample (N = 2 × 10 = 20) of 42-day-old birds were dissected on the 42nd day and are presented here. We dissected a sample of 10 birds from the XENOFOOD-fed and a sample of 10 birds from the Control groups. That is, not all but samples (N = 2 × 10) of the 2 × 34) from 42-day-old birds were dissected to get post-mortem data about a few body organs as well as about the number of *Clostridium* germs in their ilea. The body weights of these selected animals did not differ from the average of their respective (C, or T) experimental groups. After dissection, the weights of the different organs were measured.

*Statistical Analysis* ANOVA procedure was carried out by using the respective propositions of the SAS 9.4, carried out partly by Sándor Józsa and partly by András Fodor. We considered our data as an unbalanced data set. The significant differences (α = 0.05) between treatment means were assessed using the Least Significant Difference (LSD). In detail, the data analysis was performed STATJ software, Version [9.4) of the SAS System for [Windows X 64 Based Systems]; (Copyright [2013 of copyright]; SAS institute Inc. SAS, Cary, NC, USA. We used ANOVA and GLM Procedures following the requirements of the SAS 9.4 Software. The design of the experiment was a randomized complete block design with, the number of the respective treatments, concentrations, and replicates Data have been averaged to allow the analysis of variance (ANOVA). The significance of differences of the means (a. = 0.05) was determined by using t (Least Significant Difference LSD) tests or Duncan’s Multiple Range Tests, depending on the respective experiment.

## 5. Conclusions

The work presented here can be considered as a contribution to the field of application of biosynthetic non-ribosomal encoded antimicrobial peptides (NR-AMPs) controlling multi-drug resistance (MDR).Regarding the future potential application of natural compounds, several aspects have to be taken into consideration: (A) the antimicrobial potential; (B) the durability, thermotolerance, and bioavailability; (C) cytotoxicity, and another, unwanted side effects, especially if one group of target organisms are of eukaryotic (protist) pathogens.We have not found any publication concerning the comparisons of the in vitro and the in vivo effects of antimicrobial active NR-AMP molecules of EPB origin on the same eukaryote (host) organism to be protected from pathogens or parasites.Herein we demonstrated that both the heat-inactivated and the native cell-free conditioned culture media (CFCM) of *Xenorhabdus budapestensis* DSM16342 (EMA), and that of *Xenorhabdus szentirmaii* DSM16338 (EMC) which has broad-spectrum antimicrobial activity against several prokaryotic and eukaryotic microbial pathogens, also exerted a robust cytopathogenic effect on confluent (cultured) chicken cell layer.Whether cytotoxicity and antibiotic potential of the same NR-AMP molecule is an open question.The results of the in vivo experiment presented in Table 2, Table 3 and Table 4 may allow concluding that XENOFOOD feeding was harmless. It means that the NR-AMPs, present in the CFCM of *X. budapestensis* and *X. szentirmaii*-at least when applied as autoclaved food ingredients-did not cause any detectable side effects in vivo in broilers.The incorporation (by covalent chemical bounds, or complex formation) of non-decomposed NR-AMPs into proteins or other tissue components is biochemical nonsense.However, their deposition into the egg yolk cannot theoretically be ruled out, therefore, we may recommend XENOFOOD as protecting food supplement only for cockerels but not for pullets.The XENOFOOD exerted gastrointestinal antimicrobial activity in the GI has been reflected in the reduced average value of the *Clostridium* CFUs in the XENOFOOD-fed group.We suppose that the bursa-dependent humoral immune system neutralized the cytotoxic side components (of peptide nature) of the XENOFOOD, (that is, cytotoxic NR-AMPs of CFCMs in the blood (but, naturally, not in the gut).The presence of the active antimicrobial compounds in the GI is proved by the significantly reduced number of colony-forming germs in the lower jejunum of the XENOFOOD-fed (T-group of) animals compared to that of the controls, (C-group). Based on those facts, we may recommend XENOFOOD as a useful food supplement to protect broiler cockerels against coccidiosis.However, before drawing that conclusion, we recommend another experiment on experimentally *Clostridium*-pre-infested chickens, (also on cockerels, not pullet) to reproducibly determine the in vivo anti-Clostridial effects of the XENOFOOD diet quantitatively. (As mentioned above, we do not recommend the use of pullets, because the deposition of the XENOFOOD ingredients is unpredictable).

We have been looking for a cooperating partner for carrying on this final experiment, and this is one of our strongest arguments for publishing these data in *Pathogens*.

## Figures and Tables

**Figure 1 pathogens-12-00458-f001:**
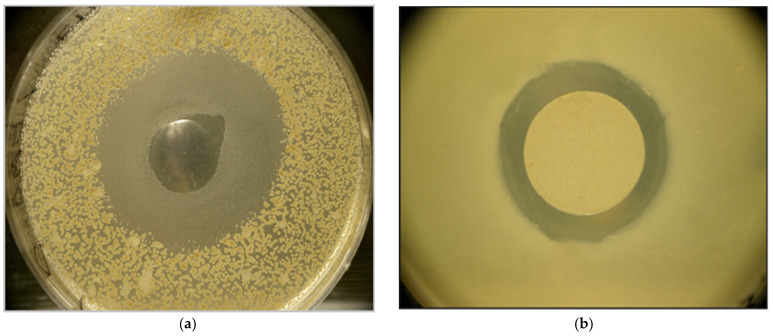
Results of the in vitro experiment with EMA CFCM in anaerobic conditions. The active ingredient inhibited the growth of *Clostridium perfringens* cells in agar media before (**a**) and after (**b**) pepsin digestion. For details, see Materials, and Method. Photo: Cs. Pintér.

**Figure 2 pathogens-12-00458-f002:**
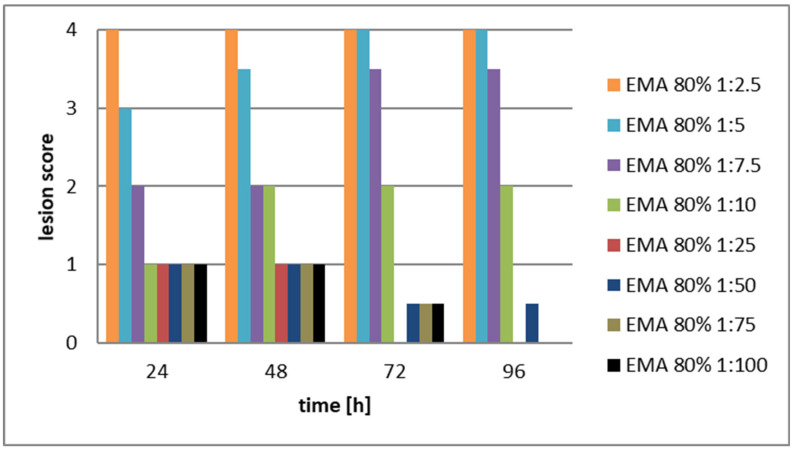
Destruction of LMH [48] cell monolayer caused by serially diluted EMA CFCM samples. The scoring system is [47] shown in Appendix A. (Abbreviations: EMA= *Xenorhabdus budapestensis* DSM16342(T) (Lengyel). CFCM= cell-free conditioned culture media). Rough data are given in Appendix A.) Above 1.5 *v*/*v*% of the EMA CFCM concentrations (that is below 1.5 *v*/*v* dilutions) the differences between the score values representing the degree of cytotoxicity differed significantly in a dose-dependent manner, at least at lower dilutions, (<8 *v*/*v*%), where the cytotoxic damages seem to be regenerating. The degree of significance was detectable at 0.001 < *p* < 0.005 level at each consecutive measurement between the following values: At 24 h: EMA 80:1:2.5 > EMA 80:1:5 > EMA 80:1:7.5 > EMA 80:1:10 = EMA 80:1:25 = EMA 80:1:50 = EMA 80:1:100. At 48 h: EMA 80:1:2.5 > EMA 80:1:5 > EMA 80:1:7.5 = EMA 80:1:10 = EMA 80:1:25 = EMA 80:1:50 = EMA 80:1:100. At 72 h: EMA 80:1:2.5 = EMA 80:5 > EMA 80:1:7.5 > EMA 80:1:10 > EMA 80:1:50 = EMA 80:1:75 EMA = 80:1:100. At 96 h: EMA 80:1:2.5 = EMA 80:5 > EMA 80:1:7.5 > EMA 80:1:10 > EMA 80:1:50.

**Figure 3 pathogens-12-00458-f003:**
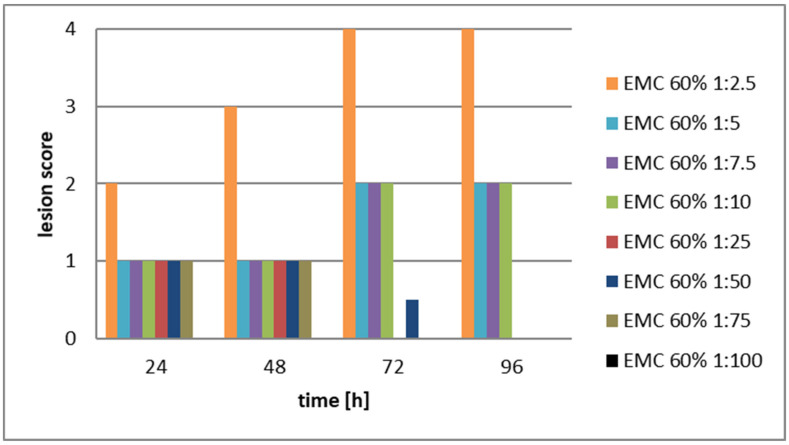
Destruction of LMH [48] cell monolayer caused by serially diluted EMC CFCM samples. The scoring system is [47] shown in Appendix A. (Abbreviations: EMC = *Xenorhabdus budapestensis* DSM16338(T) (Lengyel). CFCM = cell-free conditioned culture media). Data and statistics are given in Appendix A. EMC CFCM seems to be less cytotoxic than EMA CFCM. Above 12 *v*/*v*% of the EMC CFCM concentrations (that is below 12 *v*/*v* dilutions) the differences between the score values representing the degree of cytotoxicity did not differ significantly in a dose-dependent manner, at least at lower dilutions, (<12 *v*/*v*%), where the cytotoxic damages seem to be partially regenerating. The degree of significance was detectable at 0.001 < *p* < 0.005 level at each consecutive measurement between the following values: At 24 h: EMC 60:2.5 > EMC 60:5 = EMC 60:1:7.5 = EMC 60:1:10 = EMC 60:25 = EMC 60:1:50 = EMA = EMC 60:1:75 = EMCA 0:1:100. At 48 h: EMC 60:2.5 > EMC 60:5 = EMC 60:1:7.5 = EMC 60:1:10 = EMC 60:25 = EMC 60:1:50 = EMA = EMC 60:1:75 = EMC 0:1:100. At 72 h: EMC 60:2.5 > EMC 60:5 = EMC 60:1:7.5 = EMC 60:1:10 > EMC 60:1:50. At 96 h: EMC 60:2.5 > EMC 60:5 = EMC 60:1:7.5 = EMC 60:1:10.

**Figure 4 pathogens-12-00458-f004:**
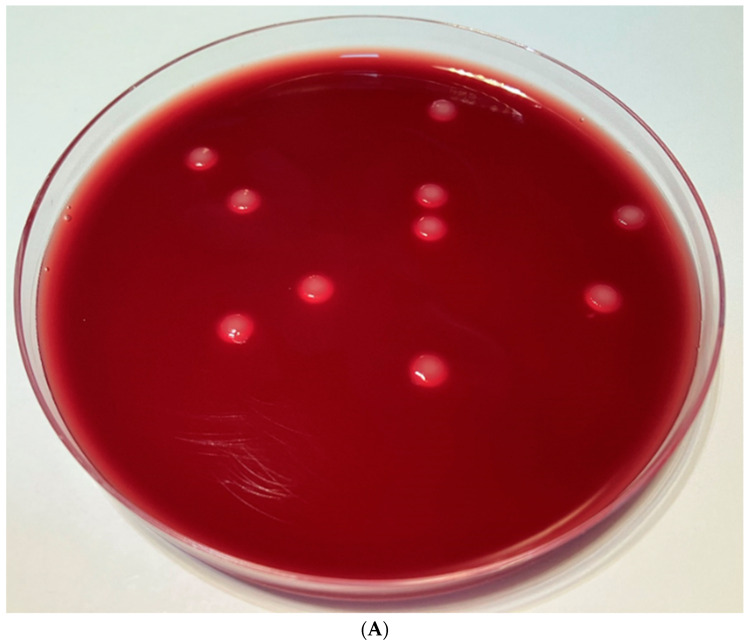
Determination of viable cell count using plate counting method (*Clostridium perfringens*, blood agar). (**A**) *Clostridium* CFU (48 h incubation). (**B**) *Clostridium* CFU (60 h incubation). (**C**) *Clostridium* CFU (72 h incubation).

**Table 1 pathogens-12-00458-t001:** Results of Experiment 1. Scoring the degree of LMH monolayer destruction caused by CFCM of 3 different EPB strains representing 2 different species.

Samples	Replicates	24 h	48 h
M199 the original (unchanged) culture Media, in which LMH layer had been developed	A	0	0
	B	0	0
	C	0	0
Fresh (199 + 15% + PKS) culture media added	A	0	0
	A		
	B	0	0
	C	0	0
EMA CFCM (EMA had been cultured inM199)	A	3	4
	B	3	4
	C	3	4
EMC CFCM (EMC had been cultured inM199)	A	4	4
	B	4	4
	C	4	4
TT01 YELLOW CFCM (TT01 had been cultured inM199)	A	4	4
	B	4	4
	C	4	4
TT01 RED CFCM TT01 had been cultured inM199)	A	4	4
	B	4	4
	C	4	4

Table 1 the numbers in the table are the score values (see Table 1). Abbreviations: M199 Unchanged C growth control media (M199, unchanged); M199 supplemented with + 15% + PKS C growth control media (fresh M199, supplemented with 15% fibroblast growth media); EMA CFCM: cell-free conditioned culture media of *Xenorhabdus budapestensis* DSM16442 (EMA); EMC CFCM: cell-free conditioned culture media of *X. szentirmaii* DSM16338 (EMC); TT01 YELLOW CFCM: cell-free conditioned culture media of *Photorhabdus luminescens* ssp. *akhurstii* TT01, yellow colony-color variant (from McConkey agar media); TT01 RED CFCM: cell-free conditioned culture media of *P. luminescens* ssp. *akhurstii* TT01 RED colony-color variant (from MacConkey agar media.

**Table 2 pathogens-12-00458-t002:** Gastro-enteral antimicrobial potential of XENOFFOD (reflected in the difference between the average numbers of the *Clostridium* CFUs in the lower ileum) of XENOFOOD-fed (T), and Control (C) group of chicken.

TREATMENTS	N	No. of *Clostridium* CFU in the Lower Ileum
C (Control)	10	149.9
T (Xenofood-fed)	10	48.1
*t*		−2.128733
*p*		0.028
Significance		

**Table 3 pathogens-12-00458-t003:** Absolute, and relative weights of immune organs (bursa of Fabricius, and spleens of XENOFOOD-fed (T) and Control (C) chicken.

Treatments		Spleen	Bursa of Fabricius		
		Weight	Weight	Size	Individual Bursa/Spleen
	N	mg	mg	mm	Ratio
C (Control)	10	2266.6	2761.0	25.2	1.39
T (Xenofood-fed)	10	1618.0	3618.2	26.9	2.38
*t*		+2.09	+3.16	3.02	3.7
*p*		0.056	0.006	0.009	0.006
Significance		NS			

Average measurements (sizes and weights) of the spleen and bursas of Fabricius.

**Table 4 pathogens-12-00458-t004:** Feed intake, weight gain, and FCR of XENOFOOD–fed chicken and control chicken.

Treatments	Growing Periods (Days)		
	Starter (1–10)	Grower (11–24)	Finisher (25–42)
Feed intake			
C (Control)	256 ± 14.9	1401 ± 43.8	3267 ± 113.3
T (XENOFOOD-fed)	267 ± 14.3	1435 ± 39.5	3376 ± 74.4
Significance	ns	ns	ns
Weight gain			
C (Control)	202.7 ± 12.5	950.4 ± 22.9	1486 ± 158.9
T (XENOFOOD-fed)	213.2 ± 10.7	975.6 ± 64.5	1600 ± 63.7
Significance	ns	ns	ns
FCR			
C (Control)	1.05 ± 0.04	1.48 ± 0.03	2.42 ± 0.40
T (XENOFOOD-fed)	1.05 ± 0.02	1.47 ± 0.08	2.26 ± 0.13
Significance	ns	ns	ns

**Table 5 pathogens-12-00458-t005:** A list of the non-ribosomal templates antimicrobial peptides discovered in different *Xenorhabdus* species.

Natural NR-AMPs	*Xenorhabus* Species	Reference
Xenofuranone A and B	*X. szetirmaii* (EMC) DSM(16338)T	Brachmann, et al., 2006 [54]
Nemaucin	*X. cabanillasii*	Gualtieri, et al., 2009a [59], Gualtieri, et al., 2012 [60]
Fabclavine	*X. budapestensis* DSM16342)T (EMA)	Fuchs et al., 2012 [43]
Fabclavine, A, B	*X. szentirmaii* DSM(16338)T (EMC)	Fuchs et al., 2014 [44]
Fabclavine, biosynthetic intermediaries, derivatives, and analogs	*X. szentirmaii* DSM(16338)T (EMC) All but e few *Xenorhabdus*	Wenski et al., 2019 [45] Wenski et al., 2020 [46]
Cabanillasin	*X. cabanillasii,* *X. khoisanae*, SB10	Houard et al., 2013 [61]
PAX peptides	*X. nematophila*	Gualtieri et al., 2009b [62]
		Fuchs et al., 2011 [63]
	*X. khoisanae*, SB10	Dreyer, et al., 2019 [64]
Odilorhabdins	*X. riobrave*	Isaacson and Webster, 2013 [65]
		Pantel, et al., 2018 [66]
		Sarciaux et al., 2018 [67]
		Racine, and Gualtieri 2019 [68]
		Lanois-Nouri, et al., 2022 [69]
Anti-oomycete peptides	*X. budapestensis* NMC-10	Xiao et al., 2012 [52]
Xenortide	*X. nematophila*	Reimer, 2014 [70]
Xenortide A-D	*X. nematophila*	Esmati, et al., 2018 [71]
Rhabdopeptide	*X. nematophila*	Reimer, et al., 2013 [72]
Rhabdopeptide (with nematicide activity)	*X. budapestensis SN84*	Bi et al., 2018 [73]
Rhabdoopeptide/xenortide-like peptides	*Xenorhabdus innexi*	Zhao, 2018 [53]
New cyclic depsipeptide xenematide F, and G, (anti-oomycete activity)	*X. budapestensis* SN84	Xi et al., 2019 [74]
Szentiamide	*X. szentirmaii* DSM16338^T^ (EMC)	Ohlendorf, et al., 2011 [56]
		Nollmann, et al., 2012 [57]
Genomic information: 71 NR-AMP operons in.	*X. szetirmaii* (EMC)^T^ DSM16338	Gualtieri et al., 2014 [58]

## Data Availability

All of the data are available from corresponding author.

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
