# Peer review of "XENOFOOD—An Autoclaved Feed Supplement Containing Autoclavable Antimicrobial Peptides—Exerts Anticoccidial GI Activity, and Causes Bursa Enlargement, but Has No Detectable Harmful Effects in Broiler Cockerels despite In Vitro Detectable Cytotoxicity on LHM Cells"

_pathogens, 2023, doi:10.3390/pathogens12030458_

Round 1

Reviewer 1 Report

 This is an interesting paper. The study by Fodor et al. suggests that XENOFOOD, an autoclaved feed supplement containing 2 autoclavable antimicrobial peptides, has anticoccidial GI activity, and causes bursa enlargement, but has no detectable  harmful effects in broiler cockerels despite in vitro cytotoxic ingredients. The cell-free conditioned culture media of Xenorhabdus budapestensis and X. szentirmaii efficiently inhibit avian pathogens like Clostridium, Histomonas, and Eimeria. XENOFOOD (containing autoclaved X. budapestensis, and X. szentirmaii cultures developed on chicken food) were consumed by the birds. The XENOFOOD exerted detectable gastrointestinal (GI) activity (reducing the numbers of the colony-forming Clostridium perfringens units in the lower jejunum. No animal was lost in their experiments, suggesting that XENOFOOD diet did not result in any detectable adverse effects. Their conclusions are not sufficiently supported by data. Several issues need to be addressed.

 1.What the real components of XENOFOOD ? The major components of

XENOFOOD could be checked by LC/MS.

2. In the induction (lines 119-123) and lines 295-298, the authors suggest that the parameters indicating a moderate enlargement of bursas of Fabricius (average weight, size, and individual bursa/spleen weight-ratios) in the XENOFOOD-fed group must be an indirect indication that the bursa-controlled humoral immune system neutralized the cytotoxic ingredients of the XENOFOOD in the blood, not allowing to reach their critical cytotoxic concentration in the sensitive tissues. The authors conclude that XENOFOOD feeding was harmless. There are no sufficient data to support their conclusion. Do you perform pathological section examination of the lesion the moderate enlargement of bursas of Fabricius ?

3.In Fig. 1, The inhibition zones of two tests are quite different.  What is the MIC? 

4. How many Gram-positive or Gram-negative bacteria could be inhibited by XENOFOOD ?  More bacterial species should be tested in this study.

5. Fig. 2-3 need statistical analysis, showing the mean± SE of three independent experiments.

6. In Table S1, what is the size of scale bar ? In Fig. 4, needs scale bar.

Reviewer 2 Report

Dear editor, this is a really good manuscript in avian reogin. The author try to learn whether a bio-preparation containing 34 antimicrobial peptides of Xenorhabdus origin with accompanying (in vitro detectable) cytotoxic 35 effects could be considered a safely applicable preventive feed supplement.

They performed excellent experiment to obtain really large number data, which show that the XENOFOOD exerted detectable gastrointestinal (GI) activity in the lower jejunum. However there were still some small mistake or unclear part, which we listed as follow.

1. Line 35 What is the antibacterial peptide produced by Xenorhabdus in the abstract?

2. The line 269-272 Fig4 picture is not clear. It is recommended to put the panorama and the partial enlarged picture on it and upload the picture with higher definition.

3. What is the bacteriostatic effect of lin 280 XENOFOOD (containing autoclaved X. 37 budapestis, and X. szentirmaii cultures developed on chicken food) on chicken? Whether it needs to be supplemented again.

Round 2

Reviewer 1 Report

The revised MS can be accepted for publication. 

Author Response

p
